# The Spectrum of Germline Nucleotide Variants in Gastric Cancer Patients in the Kyrgyz Republic

**Airat Bilyalov** [1,2,*], **Sergey Nikolaev** [2], **Anastasiia Danishevich** [2], **Igor Khatkov** [2], **Komron Makhmudov** [1], **Zhainagul Isakova** [3], **Nurbek Bakirov** [4], **Ernis Omurbaev** [4], **Alena Osipova** [2,5], **Ramaldan Ramaldanov** [4], **Elena Shagimardanova** [1], **Andrey Kiyasov** [1], **Oleg Gusev** [1,5,6] **and Natalia Bodunova** [2]

1  Institute of Fundamental Medicine and Biology, Kazan Federal University, 420008 Kazan, Russia; omgkartmen@gmail.com (K.M.); rjuka@mail.ru (E.S.); kiassov@mail.ru (A.K.); gaijin.ru@gmail.com (O.G.)
2  SBHI Moscow Clinical Scientific Center Named after Loginov MHD, 111123 Moscow, Russia; s.nikolaev@mknc.ru (S.N.); a.danishevich@mknc.ru (A.D.); i.hatkov@mknc.ru (I.K.); a.savinkova@mknc.ru (A.O.); n.bodunova@mknc.ru (N.B.)
3  Research Institute of Molecular Biology and Medicine, Bishkek 720005, Kyrgyzstan; jainagul@mail.ru
4  National Center of Oncology and Hematology of the Ministry of Health of the Kyrgyz Republic, Bishkek 720055, Kyrgyzstan; nurbek.bakirov.1977@mail.ru (N.B.); omurbaev_ernis@mail.ru (E.O.); ramaldanovramaldan1982@gmail.com (R.R.)
5  Graduate School of Medicine, Juntendo University, Tokyo 113-8421, Japan
6  Endocrinology Research Centre, 117036 Moscow, Russia
*  Correspondence: bilyalovair@yandex.ru

**Abstract:** Gastric cancer is a major challenge in modern oncology due to its high detection rate and prevalence. While sporadic cases make up the majority of gastric cancer, hereditary gastric cancer is caused by germline mutations in several genes linked to different syndromes. Thus, identifying hereditary forms of gastric cancer is considered crucial globally. A survey study using NGS-based analysis was conducted to determine the frequency of different types of hereditary gastric cancer in the yet-unstudied Kyrgyz population. The study cohort included 113 patients with diagnosed gastric cancer from Kyrgyzstan. The age of patients was 57.6 ± 8.9. Next-generation sequencing analysis of genomic DNA was performed using a custom Roche NimbleGen enrichment panel. The results showed that 6.2% (7/113) of the patients had pathogenic or likely pathogenic genetic variants. Additionally, 3.5% (4/113) of the patients carried heterozygous pathogenic/likely pathogenic variants in high penetrance genes, such as *TP53*, *POLD1*, *RET*, and *BRCA2*. Moreover, 2.7% (3/113) of the patients carried heterozygous mutations in genes linked to autosomal recessive conditions, specifically *PALB2*, *FANCA*, and *FANCD2*. We have not identified any genetic variants in hereditary GC-associated genes: *CDH1*, *STK11*, *SMAD4*, *BMPRIA*, *APC*, *MLH1*, and others. Our study included patients with sporadic features of GC. The use of recognized criteria (NCCN, Gastric Cancer, Version 2.2022) would increase the number of identified genetic variants in hereditary GC-associated genes. Further research is required to determine the clinical relevance of the genetic variants identified in the current study.

**Keywords:** gastric cancer; NGS; hereditary cancer

## 1. Introduction

Gastric cancer (GC) is one of the most pressing problems of modern oncology, due to the high frequency of detection, as well as high prevalence [1]. According to data from the World Cancer Research Fund (2019), the leading countries in terms of GC occurrence among both female and male populations are South Korea, Mongolia, and Japan. Kyrgyzstan exhibits a prevalence rate of 18.6 per 100,000 individuals, ranking 6th among the 46 Middle Eastern countries. These data serve as an example of the higher distribution of GC, although the underlying reasons for this phenomenon remain to be fully elucidated [2].

In the structure of oncological morbidity in the Kyrgyz Republic among males, GC occupies first place (15.0‰) [3]. Among women, GC ranks third in terms of prevalence

among oncological diseases (15.6‰) [4]. In the structure of mortality from oncological diseases, GC ranks first (10.0‰). Low rates of early diagnosis (17.6%) of GC, high degree of neglect (35.3%) and increased one-year mortality (81.7%) have also been noted in the article by Toigonbekov et al. [5]. According to the World Health Organization, in 2020, 1027 new cases of GC were detected in Kyrgyzstan, which is 14.5% of the total number of all cancer cases [6].

The majority of GC cases are sporadic [7]. The main causes of GC include various environmental factors such as infectious agents (*Helicobacter pylori*, *Epstein–Barr virus*), smoking, and dietary habits [8]. A family history of GC is present in 10% of cases; however, only 1–3% of all GCs are believed to be hereditary [9]. Hereditary GC is caused by germline mutations in the number of genes associated with different syndromes and characterized by an early onset of the disease and various multiple primary tumors [8]. Currently, there are three forms of hereditary GC [10]: hereditary diffuse GC (HDGC), associated with germline mutations in *CDH1* and *CTNNA1* genes; gastric adenocarcinoma and proximal polyposis of the stomach (GAPPS), caused by germline mutations in APC; and familial intestinal GC (FIGC), associated with germline variants in a number of candidate genes.

The presence of Lynch syndrome (*EPCAM*, *MLH1*, *MSH2*, *MSH6*, and *PMS2*), juvenile polyposis syndrome (*SMAD4* and *BMPR1A*) and Peutz–Jeghers syndrome (*STK11*) have been linked to elevated gastrointestinal cancer risk [11–13]. There are also other less-commonly reported hereditary cancer predisposition syndromes (HCPS) that are associated with either high or moderate GC risk, such as hereditary breast and ovarian cancer syndrome (*BRCA1/2*), Li-Fraumeni syndrome (*TP53*), Cowden syndrome (*PTEN*), Ataxia-telangiectasia (*ATM*), and Bloom syndrome (*BLM*) [10]. The overlap of phenotypes of multiple HCPS often leads to a prolonged and fruitless search for mutations in single genes. NGS-based gene panel analysis has been recognized as a preferred method due to its ability to assess the status of multiple genes simultaneously [14].

Some HCPS, such as HDGC and Lynch syndrome, have been well studied. Additionally, established criteria for patient selection, genetic testing, treatment, and prevention for GC and secondary primary cancers have been developed. However, the currently available evidence for GC screening efficiency in patients with rare HCPS is limited, thus there are no recommendations for implementing patient management [15].

Therefore, the identification of hereditary forms of GC is considered to be highly important worldwide. In Kyrgyzstan, this first survey study was conducted to determine the frequency of various hereditary GC types using NGS-based analysis.

## 2. Materials and Methods

The study included 113 GC patients observed or treated at the National Center of Oncology and Hematology of the Ministry of Health of the Kyrgyz Republic from 2017 to 2019. All patients signed an informed consent. The study was approved by the local ethics committee. All patients were enrolled for molecular genetic testing based on the inclusion criteria: patients with diagnosed GC; patients with Kyrgyz nationality; patients are not related to each other. Patients with multiple primary tumors were excluded from the study. All patients were tested for presence of *Helicobacter pylori*.

Peripheral blood samples were collected in two EDTA tubes (5 mL each) from all participants.

### 2.1. DNA Extraction

DNA was isolated from whole blood using the DNeasy Blood & Tissue Kit (QIAGEN) according to manufacturer's protocol and quantified using a Qubit 3.0 fluorometer.

### 2.2. Library Preparation and Sequencing

An amount of 100 ng of isolated DNA was used for preparation of sequencing libraries using a KAPA HyperPlus kit (Roche, Basel, Switzerland) via either enzymatic or ultrasonic fragmentation according to manufacturer's instructions. The size of the resulting library

fragments was evaluated using an Agilent 2100 Bioanalyzer (Agilent technologies, Santa Clara, CA, USA). Quantitative analysis of the final libraries was performed using the Qubit 3.0 fluorometer (Thermo Fisher Scientific, Waltham, MA, USA). The gene panel consisted of coding regions and flanking sequences of genes associated with various HCPS (Table 1).

**Table 1.** List of analyzed genes.

| |
|---|
| *SDHB, PTCH2, MUTYH, NTRK1, SDHC, CDC73, PARP1, FH, RET, BMPR1A, PTEN, SUFU, HRAS, CDKN1C, ABCC8, WT1, EXT2, SDHAF2, MEN1, MRE11A, ATM, SDHD, CDKN1B, CDK4, POLE, BRCA2, RB1, FANCM, MAX, RAD51B, MLH3, DICER1, GREM1, FANCI, BLM, NTHL1, TSC2, SLX4, PALB2, CDH1, FANCA, RPA1, TP53, FLCN, NF1, RAD51D, CDK12, ERBB2, SMARCE1, BRCA1, HOXB13, RAD51C, PPM1D, BRIP1, AXIN2, RHBDF2, RBBP8, SMAD4, STK11, SMARCA4, ERCC2, POLD1, GEN1, ALK, EPCAM, MSH2, MSH6, FANCL, TMEM127, PMS1, BARD1, SMARCB1, CHEK2, NF2, FANCD2, VHL, MLH1, BAP1, GATA2, ATR, PHOX2B, PDGFRA, KIT, FAM175A, SDHA, TERT, MSH3, APC, RAD50, CTNNA1, SPINK1, FANCE, PMS2, EGFR, RINT1, MET, POT1, PRSS1, XRCC2, PPP2R2A, RPS20, NBN, EXT1, RECQL4, CDKN2A, FANCG, FANCC, PTCH1, GALNT12, TSC1, FANCB.* |

Next-generation sequencing (NGS) was performed on the Illumina MiSeq platform using the MiSeq Reagent Kit v2 (500 cycles) (Illumina, San Diego, CA, USA). The Sanger sequencing method was utilized for the purpose of validation.

### 2.3. Variant Classification and Bioinformatics Analysis

The alignment of the paired-end fastq files to a reference sequence (hg38) was performed with the BWA-MEM2 algorithm [16]. Duplicates were marked with Picard MarkDuplicates [17]. The base quality score recalibration and variant calling were then performed using GATK BQSR and GATK HaplotypeCaller, respectively [18]. Annotation and interpretation of all identified variants were carried out using in-house pipeline and data from various databases. The clinical significance of identified variants was determined using interpretation standards and guidelines of the American College of Medical Genetics and Genomics and the Association of Molecular Pathology [19]. Additionally, the search of publications that mentioned variants identified in our study was performed using PUBMED Mastermind and VarSome databases (assessed on 6 November 2023) [20]. All the samples with mean coverage of less than 70 were excluded from the subsequent analysis.

### 3. Results

The study cohort included 113 individuals diagnosed with GC from Kyrgyzstan. The age of patients ranged from 36 to 79 years, with a mean age of $57.6 \pm 8.9$. The gender distribution was as follows: 35 females (31%) and 78 males (69%). All GC diagnoses were classified in accordance to Lauren classification. Further clinical information is presented in Table 2.

The results of our study indicate that 6.2% (7/113) patients in the cohort had pathogenic (PV) or likely pathogenic (LPV) genetic variants (Table 3). Within the group, 3.5% (4/113) of the patients carried heterozygous PV/LPV variants in high penetrance genes, such as *TP53*, *POLD1*, *RET*, and *BRCA2*. In addition, 2.7% (3/113) of the patients were found to be carriers of heterozygous mutations in genes associated with autosomal recessive conditions, specifically *PALB2*, *FANCA*, and *FANCD2*. The relative distribution of mutations is shown in Figure 1. In one patient, a combination of two heterozygous variants in the *RET* and *BRCA2* genes was identified. All patients identified above tested negative for *Helicobacter pylori*. Clinically significant variants were predominantly identified in males (6/7, 85.7%) with an average age of 64.5 years.

**Table 2.** Characteristics of the examined group.

| Characteristics | | Number of Patients (%) |
|---|---|---|
| Gender | Female | 35 (31%) |
| | Male | 78 (69%) |
| Histological types (Lauren) | Intestinal type | 77 (68.1%) |
| | Diffuse type | 21 (18.6%) |
| | Mixed type | 15 (13.3%) |
| *Helicobacter pylori* | *H. pylori +* | 16 (14.2%) |
| | *H. pylori −* | 97 (85.8%) |
| Stage of cancer | I | 5 (4.4%) |
| | II | 16 (14.2%) |
| | III | 61 (54%) |
| | IV | 31 (27.4%) |

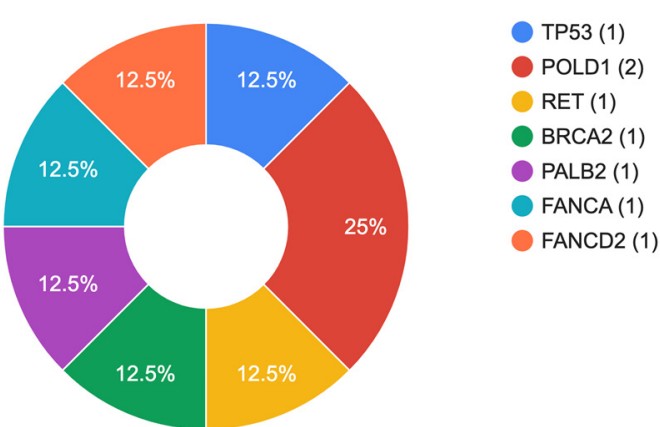

**Figure 1.** Spectrum of identified genetic variants.

Among the identified pathogenic variants, there were four (50%) frameshift, two (25%) missense, and two (25%) nonsense genetic alterations. Loss of function variants were identified in *BRCA2, POLD1, PALB2, FANCA,* and *FANCD2* genes. A total of four (50%) out of eight variants had been previously reported, while four (50%) were novel.

Our study did not reveal any genetic variants in the most common hereditary GC-associated genes: *CDH1, STK11, SMAD4, BMPRIA, APC, MLH1, MSH2,* and *MSH6*.

Several genetic alterations identified during the NGS data analysis were excluded from the study due to the fact that phenotypes of patients did not resemble the ones described in carriers of deleterious genetic variants in the genes where these alterations were located. The whole spectrum of identified genetic variants is presented in Table S1.

**Table 3.** Frequency of P/LP variants among tested individuals.

| № | Gender (Age) | Lauren Classification | Stage | Gene | Chromosomal Change | Coding | Protein | ACMG | Cancer Cases in Family History | Pathogenicity Scores * | GnomAD Exomes | Literature |
|---|---|---|---|---|---|---|---|---|---|---|---|---|
| 1 | Male (63) | D [a] | IIIB | TP53 | chr17:7675140G>A | c.472C>T | p.Arg158Cys | P [d] | No | Pathogenic-22 Uncertai-9 Benign-1 | $f$ = 0.00000796 | [21,22] |
| 2 | Male (60) | D [a] | IIIB | RET | chr10:43119548G>A | c.2410G>A | p.Val804Met | P [d] | No | Pathogenic-5 Uncertai-12 Benign-3 | $f$ = 0.000125 | [23,24] |
| | | | | BRCA2 | chr13:32340763delA | c.6410del | p.Asn2137MetfsTer31 | P [d] | | - | Not found | [25,26] |
| 3 | Female (57) | M [b] | IIIB | POLD1 | chr19:50414912delT | c.2486del | p.Leu829ArgfsTer59 | LP [e] | No | - | Not found | - |
| 4 | Male (74) | I [c] | IIIB | POLD1 | chr19:50416486G>T | c.2911G>T | p.Glu971Ter | LP [e] | Yes | Pathogenic-4 Uncertai-3 Benign-2 | Not found | - |
| 5 | Male (62) | I [c] | IIIB | PALB2 | chr16:23630456ins ACGACTT | c.1692_1698dup | p.His567LysfsTer13 | LP [e] | No | - | Not found | - |
| 6 | Male (70) | I [c] | IIIB | FANCA | chr16:89816550C>T | c.66G>A | p.Trp22Ter | P [d] | No | Uncertai-3 Benign-7 | Not found | [27–31] |
| 7 | Male (58) | D [a] | III | FANCD2 | chr3:10081422delGC | c.3182_3183del | p.Cys1061LeufsTer21 | LP [e] | No | - | Not found | - |

[a] Diffuse type; [b] Mixed type; [c] Intestinal type; [d] Pathogenic variant; [e] Likely pathogenic variant. (HG38). * Pathogenicity scores for identified genetic variants were obtained from VarSome.

## 4. Discussion

Hereditary forms of GC are predominantly observed during the third to fourth decade of life [32]. Therefore, guidelines for the diagnosis of various hereditary GCs, including those proposed by the International Gastric Cancer Linkage Consortium, recommend testing patients for hereditary GC if they manifest the disease before the age of 50 [33].

In the current study, we did not find any genetic variants in the most commonly mutated genes in GC (*CDH1*, *STK11*, *SMAD4*, *MLH1*, *MSH2*, and *MSH6*) [34]. This can be potentially explained by the lack of above-mentioned inclusion criteria, thus our study included patients with sporadic features of GC. However, we identified PV/LPV associated with other autosomal dominant and recessive conditions.

It is notable that a combination of two pathogenic heterozygous variants in *BRCA2* and *RET* genes were found in a 60-year-old patient with diffuse GC. No prior instances of cancer were recorded in the patient's family history. The neoplastic growth was situated on the lesser curvature of the stomach, in the vicinity of the cardia, transitioning into the esophagus. The first variant was c.2410G>A, located in the *RET* gene. Studies have shown that the c.2410G>A genetic variant is associated with a low to moderate lifetime risk of developing medullary thyroid cancer and multiple endocrine neoplasia 2A (OMIM#171400), but not with GC [23,24]. The *RET* (rearranged during transfection) is a well-known gene involved in the development and progression of several types of cancers, particularly medullary thyroid cancer and multiple endocrine neoplasia type 2 (MEN 2) [35]. The RET protein is a tyrosine kinase receptor that plays a crucial role in signaling pathways that regulate cell growth, differentiation, and survival [36]. Mutations in the RET gene can result in the activation of the oncogenic potential of the RET protein, thus contributing to the development of cancer. In the case of our patient, there were no specific cases of thyroid cancer in his personal and familial history.

The second variant identified in this patient was a c.6410del in the *BRCA2* gene. This variant leads to a formation of premature stop codon due to a frameshift, thus it is expected to result in an either absent or disrupted protein. Loss-of-function variants in *BRCA2* gene are known to be pathogenic [37]. This altered function can result in the loss of the protein's ability to suppress the development of cancer. The BRCA2 protein plays a crucial role in DNA damage repair, and mutations in the gene can result in increased genomic instability and a higher risk of developing cancer (OMIM #605724, #194070, #1114480, #1612555, #1613029, #155255, #613347, #176807). The c.6410del *BRCA2* genetic alteration has previously been mentioned in two publications as a variant associated with increased risk of developing breast cancer, however, there were no publications regarding its association with GC [25,26]. There have been several studies that investigated the association of *BRCA2* mutations with GC risk, but the available evidence is inconsistent. Several studies have reported a link between *BRCA2* mutations and increased risk of GC [38–40]. Further research is required in order to thoroughly comprehend the association between *BRCA2* mutations and GC risk.

The *POLD1* gene encodes a protein, which is a critical component of the DNA replication machinery and plays a crucial role in maintaining the stability of the genome [41]. The *POLD1* (Polymerase (DNA-directed), delta 1) gene is a critical component of the DNA replication machinery and plays a crucial role in maintaining the stability of the genome [41]. Mutations in the *POLD1* gene have been associated with several genetic disorders, including autosomal dominant inherited forms of neurodegeneration, such as neuroferritinopathy, as well as certain forms of colorectal and endometrial cancers [42–44].

In our study, we have identified two loss-of-function variants in *POLD1* gene: c.2486del and c.2911G>T. The former was identified in a 57-year-old female patient with mixed-type GC. She did not have any cancer in her family history. The tumor was located on both curvatures of the stomach. The latter genetic variant was identified in a 74-year-old male patient, who was diagnosed with an intestinal-type GC. The neoplasm was situated on the posterior wall of the stomach near the small curvature. The patient had a familial history of cancer: his brother died due to complications of advanced GC at the age of 47.

The c.2486del and c.2911G>T *POLD1* variants were not found in any published studies according to the genomic databases and are not present in the gnomAD genomes. These variants are located in exons 20 and 23 of the *POLD1* gene, respectively, and expected to result in a truncated non-functional protein due to formation of a premature stop codon. In a study conducted by Zhu et al., 613 GC samples were analyzed and the total frequency of *POLD1* mutations was found to be 2.7% (17 out of 613 patients) [45].

Additionally, we have identified a c.472C>T *TP53* variant in a 63-year-old male patient with diffuse GC and absence of cancer cases in the family history. The tumor was located in the stomach antrum. Mutations in the *TP53* gene are among the most common genetic alterations found in human cancer [46]. PV and LPV can disrupt the normal function of the p53 protein, leading to the uncontrolled growth of cells and the development of cancer [47]. Somatic *TP53* mutations have been found in many types of cancer, including but not limited to: lung cancer, breast cancer, ovarian cancer, colorectal cancer, and GC [48]. There are no documented cases of the germline or somatic variant c.472C>T *TP53* being associated with GC to date. Nevertheless, the literature suggests that this variant in somatic form may be linked to other cancers, such as squamous cell carcinomas, pancreatic cancer, and colorectal carcinoma [21,22]. Tan P. and Yeoh K.G. reported that approximately 50% of GC cases in the Asia-Pacific region were characterized by *TP53* somatic mutations [49]. Heterozygous germline *TP53* alterations are associated with the Li–Fraumeni syndrome [50]. The c.472C>T variant has previously been described in individuals with clinical features of Li–Fraumeni syndrome [51–55]. According to several studies, c.472C>T is expected to disrupt the function of TP53; however, numerous experimental studies have reported opposing evidence on the deleterious effect of this variant on the protein's function [56–59]. The manifestation of early-onset GC has been observed as a constituent of Li–Fraumeni syndrome, indicating the requirement of timely and recurring endoscopic screening among individuals with germline TP53 mutations, especially in those with a family history of GC [60].

A c.1692_1698dup variant in the PALB2 gene was identified in a 62-year-old male patient with intestinal type of GC. The neoplasm was localized positioned on the posterior wall of the stomach near the small curvature, in the proximal region adjacent to the esophagus. There was no reported family history of cancer. c.1692_1698dup is located in the 5th exon of *PALB2* gene and leads to a formation of a preliminary stop codon as a result of a frameshift, potentially leading to a truncated protein product. This variant has not been previously described, thus its effect on the PALB2 protein is not known. PALB2 (partner and localizer of BRCA2) is a protein encoded by the eponymous gene that plays a critical role in the process of DNA damage repair [61]. Biallelic mutations in the PALB2 are known to cause Fanconi anemia of complementation group N (OMIM# 610832). Heterozygous mutations in the PALB2 gene have been previously linked to susceptibilities to breast cancer (OMIM#114480) and GC. No records of the c.1692_1698dup PALB2 variant were found in the genomic databases. The study conducted by Fewings et al. provided evidence for the involvement of PALB2 in hereditary GC susceptibility [62]. The results indicated that individuals carrying the PALB2 mutation have an elevated risk of developing GC, especially those with a family history of the disease. Furthermore, individuals with both the PALB2 mutation and a familial history of GC are at an even higher risk and are likely to develop the disease at an earlier age [63].

Fanconi anemia (FA) is a rare genetic disorder characterized by bone marrow failure, developmental abnormalities, and an increased risk of cancer. FA is caused by mutations in any of the genes involved in the FA pathway, a complex DNA repair pathway that is critical for maintaining genome stability. In the current study, two loss-of-function variants were identified in the key genes of this pathway: *FANCA* and *FANCD2*. A 70-year-old male patient with intestinal type of GC was found to be a carrier of the c.66G>A variant in the *FANCA* gene. The neoplasm was located in the distal region of the stomach, encompassing both curvatures. No familial history of cancer was reported. The *FANCA* gene, located on chromosome 16q24.3, encodes the Fanconi anemia complementation group A protein [64].

This protein is a crucial component of the Fanconi anemia (FA) pathway, which plays a critical role in responding to DNA damage and preserving genomic stability [65]. Biallelic mutations in the *FANCA* gene, either in a compound heterozygous or homozygous form, are a cause of Fanconi anemia (#227650), which is characterized by chromosomal instability, bone marrow failure, and increased susceptibility to cancer. Studies have indicated that individuals with Fanconi anemia have an elevated risk of developing various types of cancer, including acute myeloid leukemia, squamous cell carcinoma of the head and neck, and gastrointestinal cancer. In the genomic database, we found 18 articles on the c.66G>A *FANCA* genetic variant. None of them mentioned any link of this variant to GC. Out of these articles, thirteen reported c.66G>A in homozygous form as part of Fanconi anemia, and five reported heterozygous form of this variant in patients with various cancers such as breast cancer, prostate cancer, hepatocellular carcinoma, and pancreatic ductal adenocarcinomas [27–31].

One of our patients carried the c.3182_3183del variant in another gene from FA pathway: *FANCD2*. This variant has not previously been reported in any of the published studies. However, this variant was identified in a 58-year-old patient diagnosed with diffuse GC localized in the antrum of the stomach. No family history of cancer was reported. The *FANCD2* gene, located on chromosome 3p25.3, encodes a protein that is monoubiquitinated by the FA core complex in response to DNA damage [66]. Monoubiquitination of FANCD2 leads to its recruitment to sites of DNA damage and the assembly of a multi-protein complex that is involved in DNA repair. FANCD2 also interacts with other proteins involved in DNA repair, such as BRCA1 and BRCA2 [67]. Mutations in FANCD2 are less common than mutations in *FANCA*, accounting for approximately 10–15% of FA cases [68]. Individuals with *FANCD2* mutations typically present with a milder form of FA, characterized by a later onset of bone marrow failure and a lower risk of cancer [69].

Mutations in the *FANCA* and *FANCD2* genes can increase the risk of gastric cancer by disrupting the DNA damage response pathway [70,71]. Defective DNA repair, caused by mutations in these genes, can lead to genomic instability and mutations in other genes that contribute to cancer development. FANCA and FANCD2 are also involved in scavenging reactive oxygen species (ROS), and mutations in either gene can compromise this ROS defense system, leading to increased vulnerability to ROS-induced DNA damage [72,73]. Epigenetic changes, such as alterations in DNA methylation and histone acetylation, may also contribute to gastric cancer development in individuals with mutations in these genes. The molecular mechanisms of gastric cancer development in *FANCA* and *FANCD2* mutations are complex and require further research to fully understand. Identification of these mechanisms may lead to potential therapeutic targets for the prevention and treatment of gastric cancer.

## 5. Conclusions

It is important to note that there is limited information available on the prevalence of hereditary forms of GC in the Kyrgyz Republic. Our results indicate that the prevalence of hereditary GC cases within the Kyrgyz population is similar to global levels. Further research with specific inclusion criteria in this population is necessary to gain a better understanding of germline variants' contribution to the development of gastric cancer in this region. Such studies would aid the development of more efficient and personalized screening and treatment strategies that would be based on the genetic testing results. Improving access to genetic counseling and testing services for individuals and families with a suspected or confirmed hereditary gastric cancer can facilitate the reduction of disease burden and provide opportunities for early diagnosis and intervention in the Kyrgyz Republic.

**Supplementary Materials:** The following supporting information can be downloaded at: https://www.mdpi.com/article/10.3390/cimb45080403/s1. Table S1: The whole spectrum of identified genetic variants.

**Author Contributions:** Conceptualization, A.B., S.N. and A.D.; methodology, A.D., S.N., Z.I., N.B. (Nurbek Bakirovand), E.O. and R.R.; validation, A.O.; formal analysis, A.D., S.N. and K.M.; investigation, A.B., S.N. and A.D.; data curation, S.N. and A.D.; writing—original draft preparation, A.B., S.N. and A.D; writing—review and editing, A.B., S.N., A.D., E.S. and O.G.; supervision, I.K., A.K., O.G. and N.B. (Natalia Bodunova); project administration, O.G. and N.B. (Natalia Bodunova) All authors have read and agreed to the published version of the manuscript.

**Funding:** This research received no external funding.

**Institutional Review Board Statement:** Ethics committee name: The Local Ethics Committee of the Moscow Clinical Scientific Center, named after A. S. Loginov. Approval code: 46756/15.1. Approval date: 3 February 2022.

**Informed Consent Statement:** Informed consent was obtained from all subjects involved in the study.

**Data Availability Statement:** The data are not publicly available because they contain information that could compromise the privacy of the research participants. Requests to access the additional data should be addressed to the following email: s.nikolaev@mknc.ru.

**Acknowledgments:** This paper has been supported by the Kazan Federal University Strategic Academic Leadership Program (PRIORITY-2030).

**Conflicts of Interest:** The authors declare no conflict of interest.

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
