# Peer review of "The Spectrum of Germline Nucleotide Variants in Gastric Cancer Patients in the Kyrgyz Republic"

_cimb, doi:10.3390/cimb45080403_

Round 1

Reviewer 1 Report

The authors have attempted to identify the spectrum of germline nucleotide variants in gastric cancer patients in Kyrgyz Republic.

Please cite reference in the line "The majority of GC cases are sporadic."

Please cite reference for the line "....Peutz-jeghers Syndrome (STK11) have been linked to elevated gastrointestinal cancer risk.."

What explanations would you suggest for identifying variations in only 7 out of 113 patients?

Patients with heterozygous recessive variants might not be causal, any comments?

Is this low number of patients with disease-causing mutations because patients might not have genetic history, as this was one of the shortlisting criteria.

What is the allele frequencies of the identified variations in the different global population?

Why only list of genes were selected for analysis?

Author Response

Dear Dr./Prof. Reviewer,

Thank you for reviewing our manuscript entitled "The Spectrum of Germline Nucleotide Variants in Gastric Cancer Patients in Kyrgyz Republic," submitted to Current Issues in Molecular Biology. We appreciate your time and valuable feedback on our work. We have carefully considered all of your comments and suggestions, and we would like to respond to each of them individually.

  1. Please cite reference in the line "The majority of GC cases are sporadic." - Done.

  1. Please cite reference for the line "....Peutz-jeghers Syndrome (STK11) have been linked to elevated gastrointestinal cancer risk.." – Done.

  1. What explanations would you suggest for identifying variations in only 7 out of 113 patients? - According to global data, hereditary forms of cancer are detected before the age of 50. However, the focus of our research is general population screening, which does not rely on specific criteria. As a result, the frequency of clinically significant mutations was found to be low.

  1. Patients with heterozygous recessive variants might not be causal, any comments? - In our study, two heterozygous variants were identified in FANCA and FANCD2 genes. These genes are associated with recessive conditions, thus absence of either second mutation in compound heteozygous form or hemizygous state of mutation rules out the causal character of the identified genetic variant. Due to the fact, that phenotype associated with these genes wasn’t observed in the patients, we haven’t proceeded with extended testing (e.g. WGS) with the aim of identifying other clinically significant variants in these genes.

  1. Is this low number of patients with disease-causing mutations because patients might not have genetic history, as this was one of the shortlisting criteria. - The absence of a family history of cancer in the inclusion criteria could reduce the number of identified variants in this cohort of patients.

  1. What is the allele frequencies of the identified variations in the different global population? – We included gnomAD exomes frequencies to the Table 2.

  1. Why only list of genes were selected for analysis? - The design of this panel was intended to assess the carrier status in genes, associated with the most prevalent and extensively studied hereditary cancer syndromes (according to NCCN guidelines and literature).

Once again, we sincerely appreciate your time and effort in reviewing our manuscript. We believe that the revised version of our work now meets the high standards of Current Issues in Molecular Biology. We hope that you find the changes satisfactory, and we remain open to any further suggestions or feedback you may have.

Thank you again for your valuable input. We look forward to hearing from you soon.

Yours sincerely,

Dr. Airat Bilyalov, Moscow Clinical Scientific Center named after Loginov MHD.

BilyalovAir@yandex.ru

+7 (962) 560 38 02

Reviewer 2 Report

The authors employ a custom-based next generation sequencing enrichment panel to discern germline variants in gastric cancer patients of Kyrgyz Republic. 

Strengths:  The work happens to be  a unique-centric study and perhaps a first attempt to find such variants in the subpopulation.

Weaknesses/Limitations: The germline mutations from the database an dhow of methodology was largely missing 

The bioinformatics analyses and population of databases like Varsome, CADD/GERP etc and rigorous statistics including the minor allele frequencies, Depth of coverage is missing. 

Are there any other hereditary informatin available from other related cancers from sub-population that could have allowed authors to document the study?

The inhouse pipeline and data availability statement must be accounted for.

 Minor but essential 

A pictorial methodology would be a nice addition

Helicobacter pylori must be in italics

I made subtle changes, pl find attached

The methodology is very weak 

Author Response

Dear Reviewer,

We would like to express our gratitude for your careful evaluation of our manuscript titled "The Spectrum of Germline Nucleotide Variants in Gastric Cancer Patients in Kyrgyz Republic," which we have submitted to Current Issues in Molecular Biology. We appreciate your time and effort in providing us with your insightful comments and suggestions. In response, we have made several revisions to address the issues raised.

  1. The bioinformatics analyses and population of databases like Varsome, CADD/GERP etc and rigorous statistics including the minor allele frequencies, Depth of coverage is missing  We included gnomAD exomes frequencies and in silico predictors to the Table 2. Depth of coverage was included to material and methods.
  2. Are there any other hereditary informatin available from other related cancers from sub-population that could have allowed authors to document the study? Unfortunately, we do not have such information, because this manuscript demonstrates the first population screening for carriers of germinal variants in patients with gastric cancer in the Republic of Kyrgyzstan. In the future we plan to conduct a study that is going to include specific selection criteria, higher number of gastric cancer patients with more detailed clinical information.
  3. The inhouse pipeline and data availability statement must be accounted for - Data Availability Statement was written in line 438.
  4. Minor but essential  - Done.

Once again, we sincerely appreciate your thoughtful evaluation of our manuscript. Thank you very much for your subtle changes, we included there to the manuscript. We believe that the revisions we have made in response to your comments have significantly improved the quality and clarity of our work. We hope that you find the revised manuscript satisfactory for publication in Current Issues in Molecular Biology. We are grateful for your continued consideration and look forward to receiving your final decision.

Thank you and best regards,

Yours sincerely,

Dr. Airat Bilyalov, Moscow Clinical Scientific Center named after Loginov MHD.

BilyalovAir@yandex.ru

+7 (962) 560 38 02

Round 2

Reviewer 1 Report

No Comments.

Reviewer 2 Report

Thank you.  I am satisfied with the changes.  A few more here which can be done during proofs.

L143:  mutations ARE shown

L144; WERE identified